# Contrast-Enhanced Liver Magnetic Resonance Image Synthesis Using Gradient Regularized Multi-Modal Multi-Discrimination Sparse Attention Fusion GAN

**DOI:** 10.3390/cancers15143544

**Published:** 2023-07-08

**Authors:** Changzhe Jiao, Diane Ling, Shelly Bian, April Vassantachart, Karen Cheng, Shahil Mehta, Derrick Lock, Zhenyu Zhu, Mary Feng, Horatio Thomas, Jessica E. Scholey, Ke Sheng, Zhaoyang Fan, Wensha Yang

**Affiliations:** 1Department of Radiation Oncology, Keck School of Medicine of USC, Los Angeles, CA 90033, USAapril.vassantachart@med.usc.edu (A.V.); shahil.mehta@med.usc.edu (S.M.);; 2Department of Radiation Oncology, UC San Francisco, San Francisco, CA 94143, USA; 3Guangzhou Institute of Technology, Xidian University, Guangzhou 510555, China; 21181214217@stu.xidian.edu.cn; 4Department of Radiology, Keck School of Medicine of USC, Los Angeles, CA 90033, USA

**Keywords:** MR synthesis, GAN, multi-modal fusion, tumor monitoring, contrast enhancement

## Abstract

**Simple Summary:**

Contrast-enhanced MR has been used in diagnosing and treating liver patients. Recently, development in MR-guided radiation therapy calls for daily contrast MR for tumor targeting. However, frequent contrast injection is risky to patients. We developed a deep learning model (GRMM-GAN) to synthesize contrast-enhanced MR from pre-contrast images. GRMM-GAN adopts gradient regularization and multi-discrimination mechanisms. It shows superior performance compared with state-of-the-art deep learning models.

**Abstract:**

Purposes: To provide abdominal contrast-enhanced MR image synthesis, we developed an gradient regularized multi-modal multi-discrimination sparse attention fusion generative adversarial network (GRMM-GAN) to avoid repeated contrast injections to patients and facilitate adaptive monitoring. Methods: With IRB approval, 165 abdominal MR studies from 61 liver cancer patients were retrospectively solicited from our institutional database. Each study included T2, T1 pre-contrast (T1pre), and T1 contrast-enhanced (T1ce) images. The GRMM-GAN synthesis pipeline consists of a sparse attention fusion network, an image gradient regularizer (GR), and a generative adversarial network with multi-discrimination. The studies were randomly divided into 115 for training, 20 for validation, and 30 for testing. The two pre-contrast MR modalities, T2 and T1pre images, were adopted as inputs in the training phase. The T1ce image at the portal venous phase was used as an output. The synthesized T1ce images were compared with the ground truth T1ce images. The evaluation metrics include peak signal-to-noise ratio (PSNR), structural similarity index (SSIM), and mean squared error (MSE). A Turing test and experts’ contours evaluated the image synthesis quality. Results: The proposed GRMM-GAN model achieved a PSNR of 28.56, an SSIM of 0.869, and an MSE of 83.27. The proposed model showed statistically significant improvements in all metrics tested with *p*-values < 0.05 over the state-of-the-art model comparisons. The average Turing test score was 52.33%, which is close to random guessing, supporting the model’s effectiveness for clinical application. In the tumor-specific region analysis, the average tumor contrast-to-noise ratio (CNR) of the synthesized MR images was not statistically significant from the real MR images. The average DICE from real vs. synthetic images was 0.90 compared to the inter-operator DICE of 0.91. Conclusion: We demonstrated the function of a novel multi-modal MR image synthesis neural network GRMM-GAN for T1ce MR synthesis based on pre-contrast T1 and T2 MR images. GRMM-GAN shows promise for avoiding repeated contrast injections during radiation therapy treatment.

## 1. Introduction

The American Cancer Society has estimated 41,260 newly diagnosed liver cancer and intrahepatic bile duct cancer patients in the United States and 30,520 related deaths [1]. Medical imaging is essential in both the diagnosis and treatment of liver cancer. Magnetic resonance (MR) imaging provides excellent soft tissue contrast with its versatile and functional imaging sequences. Compared to computed tomography (CT), multi-phase contrast-enhanced MR has shown improved sensitivity in detecting hepatocellular carcinoma (HCC), the most common type of primary liver cancer [2]. Hence, several guidelines recommend multi-phase contrast-enhanced MR as the standard imaging modality for liver cancer diagnosis [3,4,5,6]. Treatments of liver cancer include surgery, liver transplant, thermal ablation, chemo-radioembolization, external beam radiation therapy (EBRT), targeted drug therapy, or immunotherapy. EBRT has the advantage of being geometrically targeted and non-invasive within these modalities. However, the outcome of conventional fractionated EBRT is unsatisfactory due to tumor radio-resistance and the risk of radiation-induced liver disease [7]. Stereotactic body radiation therapy (SBRT), a more precise, ablative type of radiation, has gained popularity in the recent decade in overcoming the radio-resistance of various cancers, including liver cancer [8]. SBRT is an aggressive form of EBRT delivering highly hypo fractionated, thus more biologically potent, doses. The success of SBRT hinges on geometrically accurate tumor targeting and rapid dose drop-off to spare the surrounding normal tissues. In other words, SBRT requires a more stringent tumor definition and reduced geometrical margin. To this end, multi-phase contrast-enhanced MR has been increasingly used to register with CT for SBRT planning to better define tumor and normal anatomy interfaces. However, deformable image registration between CT and MR can be error-prone [9].

More recently, MR-guided linear accelerators (LINACs) have been commercialized, making MR-only SBRT planning and adaptation an appealing alternative circumventing MR-CT registration. Besides being more sensitive in detecting liver tumors for more accurate treatment, contrast-enhanced MR imaging could be used for daily treatment response assessment by providing better visibility and richer details in the region of interest as the contrast agents flow with blood vessels over time. However, obtaining daily contrast-enhanced images on MR-LINAC can be prohibitive due to the increased risk of side effects from repeated contrast injections [10,11,12]. The side effects can be severe for patients with compromised kidney function, a condition more commonly observed in patients with liver cancer [13]. Safety concerns regarding administering gadolinium-based contrast agents, nephrogenic systemic fibrosis, and additional procedure time [14,15,16] also preclude more frequent contrast-enhanced MR acquisition. The limitation on contrast usage thus severely diminishes the value of contrast-enhanced MR in MR-guided radiotherapy for daily tumor targeting and early response assessment. Therefore, there is a clinical need for predicting contrast-enhancement MR information without repeated contrast injection.

Medical image synthesis [17] is a rapidly developing area benefitting from deep learning (DL) methods, among which the generative adversarial network (GAN) [18,19] was mainly designed for image synthesis. The state-of-the-art GAN methods include pix2pix [20], which enforced the L1 norm paired image similarity and was efficient in paired image synthesis, and CycleGan [21], which learned the high-level features in the source domain and then applied them for style translation. GAN medical imaging contributions include medical image enhancement [22,23,24], super-resolution [25,26], cross-modal MR-CT image synthesis [27,28], and multi-contrast MR synthesis [29,30,31].

Multi-modal learning has attracted increasing interest in MR image synthesis due to the available salient and mutual complementary information. However, inherent redundancy and noisy artifacts exist across multiple modalities, making the efficacious fused learning from multi-modal MR images difficult. Several studies attempted to tackle the challenge. MM-Syns [32] learned a shared latent feature space for multi-modal data and synthesized the multi-output brain an MR image via an encoder–decoder structure. Similarly, LR-cGAN [33] adopted the encoder–decoder design in a GAN-based model for brain MR image synthesis. Hi-Net [34] proposed a mixed fusion block with element-wise operations for multi-modal feature fusion and brain MR image synthesis. MMgSN-Net [35] adopted the structure of Hi-Net with improved self-attention fusion for nasopharyngeal carcinoma MR image synthesis. MustGAN [36] designed a flexible multi-stream fusion framework to aggregate information across multiple sources and predicted missing modality data.

However, abdominal contrast-enhanced MR image synthesis from multi-modal inputs (e.g., T2, T1pre) is still challenging. To the best of our knowledge, there is currently no reported work for multi-modal contrast-enhanced liver MR image synthesis. We hypothesize that three difficulties account for this gap: (1) the inherent heterogeneity across different tumor types and patients in abdominal MR images. Compared with MR image synthesis for other anatomical sites, there is more significant variation in abdominal MR image characteristics due to different tissue composition, vascularization, perfusion, and motion artifacts. The interpatient heterogeneity poses substantial challenges to the discriminative feature learning of multi-modal fusion methods. (2) Co-registration error across abdominal MR modalities. Inevitable breathing motion during the collection of abdominal MR images results in mismatches between different sequences. The uncertainties in the co-registration could confuse a synthesis method, especially for the deep learning model, because of strong memory. (3) Blurring and over-smoothing effect of MR image synthesis. The blurring and over-smoothing effects of synthesized images are common and recognized as an issue in GAN-based models [37,38].

This study presents a novel image gradient-regularized multi-modal multi-discrimination sparse attention fusion generative adversarial network (GRMM-GAN), a non-intrusive, efficient, and cost-saving clinical tool for contrast-enhanced abdominal MR image synthesis. The GRMM-GAN produces synthetic contrast-enhanced abdominal MR images to enable more accurate tumor delineation and response assessment for adaptive liver radiation therapy.

## 2. Methods

### 2.1. Data and Preprocessing

With IRB (21-33858) approval, we randomly solicited 165 MR studies of 61 liver patients from our institutional database. The patients’ demographic and clinical information is summarized in Table 1. Each study included three modalities, T2, T1 pre-contrast (T1pre), and T1 contrast-enhanced (T1ce) at the portal venous phase. The portal venous phase was used to test the MR synthesis idea. In theory, different GRMM-GAN can be built for the arterial and delayed phases. The three modalities of each study were rigidly registered in VelocityAI^TM^ (Varian, a Siemens Healthiness company, Palo Alto, CA, USA). Various resolutions, including 640 × 640, 512 × 512, and 320 × 320, were used in the original MR images. In practice, we downsampled the MR images to 256 × 256 to balance computational cost and image quality.

### 2.2. Overall Pipeline

The GRMM-GAN MR synthesis pipeline consists of a conditional GAN baseline and three modules: sparse attention fusion, gradient regularization (GR) mechanism, and multi-scale multi-smoothness discriminators (MMD), as shown in Figure 1a. Specifically, the sparse attention fusion module was designed to extract modal-specific features from each input modality (T1pre and T2) while eliminating the inherent redundancy and noisy artifacts and discovering the salient and mutual complementary information. The image gradient regularization mechanism preserves the crucial texture and edge features of the abdominal organs and blood vessels such that the inherent heterogeneous knowledge of the abdominal MRs can be well studied. The multi-scale multi-smoothness discriminators examine the synthesis images in different scales and smoothness levels to overcome the blurring and over-smoothing effect. The following sections explain the baseline model and each network module in detail.

### 2.3. Baseline Conditional GAN Model

The GRMM-GAN method is derived from a conditional GAN framework consisting of a generator G and a discriminator D. The generator’s objective G is to fuse the multi-modal pre-contrast MR inputs, T1pre and T2, and predict the contrast-enhanced T1ce. At the same time, the discriminator D focuses on the discrimination of the synthetic MR from the real MR. The prime conditional GAN objective function $L_{cGAN}$ is given by:(1)$$L_{cGAN}(G,D)={Ε}_{x_{1},x_{2},y}\left[\log D(x_{1},x_{2},y)\right]+{Ε}_{x_{1},x_{2}}\left[\log\left(1-D\left(x_{1},x_{2},G\left(x_{1},x_{2}\right)\right)\right)\right]$$
where the multi-modal inputs and target reference are denoted as $x_{1},\;x_{2}$, and y, respectively.

Following the pix2pix method [20], the L1 loss is employed to promote the structural consistency between the real and the synthetic output and to avoid blurring. The L1 loss is defined as:(2)$$L_{L1}(G)={Ε}_{x_{1},x_{2},y}\left[{\left\|y-G\left(x_{1},x_{2}\right)\right\|}_{1}\right]$$

### 2.4. Sparse Attention Fusion

Although the current conditional GAN-based method shows promise for MR image synthesis, the inherent redundancy and noisy artifacts across multiple modalities present challenges for the effective fused learning of multi-modal abdominal MR images. To achieve superior fusion by discovering the salient and mutual complementary information in multiple MR modalities while discriminating redundancy and noise from distortion and ambiguous co-registration, we designed a sparse attention fusion network, as shown in Figure 1b. This module was inspired by the previously proposed L_1_ sparsity-regularized attention feature fusion work (L1-attention fusion) [39]. Specifically, a sparse regularization term $\lambda \left\|c_{i}\right\|$ was introduced to the learned attention ($c_{i}$) where the sparsity regularizer (λ) controls the sparsity level of the attention weights. The sparse attention updates were realized using soft thresholding after the forward inference at each iteration, shown as follows:(3)$$c_{}^{*}=S_{\lambda }(c),\;\;s.t.\;S_{\lambda }(c)=\max(c-\lambda ,0)$$

The sparsity constraint, applied to the attention estimated for multi-modal fusion, eliminates the inherent redundancy across modalities and improves discriminative ability.

### 2.5. Gradient Regularization Mechanism

We added an image gradient regularization to manage the inter-patient heterogeneity and capture salient high-level features. At the same time, the network discriminated the redundancy and artifacts from the noisy inputs. The gradient regularization mechanism [40] introduced a gradient preservation loss, shown as the following:(4)$$L_{GR}(G)={\left\|g⊗G(x_{1},x_{2})-g⊗y\right\|}_{2}^{2}={\left\|g⊗(G(x_{1},x_{2})-y)\right\|}_{2}^{2}$$
where g is the Sobel gradient operator [41], and ⊗ denotes the convolution operation, as shown in Figure 2a. A similar idea was recently applied as Ea-GANs for brain MR image synthesis [42]. The gradient regularization mechanism enforced the fidelity of the first-order information between the synthetic and real T1ce images. It thus preserved the crucial texture and edge features of the abdominal organs and blood vessels while increasing the module robustness to heterogeneous redundancy and noise. Figure 2a shows resultant gradient maps for the synthetic and real T1ce images after the Sobel operation.

### 2.6. Multi-Discrimination Mechanism

In GAN-based models for image synthesis, blurring and over-smoothing effects are commonly observed since the optimization of the fidelity loss could easily fall on a local optimum [37,38]. This problem becomes more severe and visible due to the heterogeneous nature and ambiguous co-registration of multi-modal abdominal MR images. Inspired by the multi-scale discriminators [43,44] that enhanced the resolution of the image synthesis, as well as the illumination and scale-invariant SIFT feature learning [45,46], we propose multi-scale multi-smoothness discriminators (MMD) to improve the discriminative ability and counter over-smoothness in synthesized images. The multi-modal inputs, synthesis, and ground truth images were downsampled twice into half of their original size. Then a Gaussian filter was applied to blur the synthetic image to simulate the over-smoothness effect. In total, there were three scales (256′256, 128′128, and 64′64), creating two pairs each with the original image downsampled over smoothed image for the discriminator. Three discriminators were trained given the paired blurred synthetic and real images, as shown in Figure 2b. The module thus learned blurring and over-smoothing effects in the synthesis and enforced the discrimination against these effects in adversarial training. The multi-scale multi-smoothness discrimination function is given as follows:(5)$$\underset{G}{min}\underset{D_{1},D_{2},D_{3}}{max}\sum_{k=1,2,3}L_{cGAN}(G,D_{k}).$$
where *G* is the generator for image synthesis, $D_{k},k=1,2,3$ are the three discriminators for over-smoothing discrimination, and $L_{GAN}$ is the conditional *GAN* loss function.

### 2.7. Objective and Optimization

After incorporating the sparse attention fusion, image gradient regularization, and the multi-scale multi-smoothness discriminators into the prime conditional GAN model, we wrote the final objective of the proposed GRMM-GAN formula as follows:(6)$$\underset{G}{min}\left(\left(\underset{D_{1},D_{2},D_{3}}{max}\sum_{k=1,2,3}L_{cGAN}(G,D_{k})\right)+\lambda_{1}L_{GR}(G)+\lambda_{2}L_{L_{1}}(G)\right).$$
where $\lambda_{1}$ and $\lambda_{2}$ are the trade-off hyper-parameters to balance the influence of different loss terms.

The proposed network was implemented on a workstation with Intel i9 7900x CPU and NVIDIA RTX 2080Ti´4 GPU under the PyTorch 1.4 and Ubuntu 18.04 environment. For the contrast-enhanced MR image synthesis task, the training procedure of our proposed model took 150 epochs, and we applied the Adam optimization algorithm with a batch size of 16 to update the network parameters. The balance hyper-parameter $\lambda_{2}$ was set to 100. $\lambda_{1}$ took a linear increase from 0 to 50 in the first 50 epochs and was frozen to 50 afterward. The learning rate for the first 50 epochs was fixed at 0.0002 and then linearly decreased to 0 in the following 100 epochs.

## 3. Model Evaluation

The performance of the proposed GRMM-GAN was evaluated quantitatively and compared with four state-of-the-art image synthesis networks: pix2pix [20], LR-cGAN [33], Hi-Net [34], and MMgSN-Net [35]. Pix2pix is the widely applied single modal image-to-image translation model in which both T2 and T1pre were set as the input. The other three comparisons are all state-of-the-art multi-modal MR image synthesis methods. In addition, 6 radiation oncologists (RadOnc 1–6), including 2 board-certified and 4 residents, performed Turing [47] tests to determine the authenticity of the synthesized T1ce images against the real images on 100 randomly (1:1 ratio) selected axial slices.

Three widely applied statistics metrics for evaluating the medical imaging synthesis include peak signal-to-noise ratio (PSNR), the structural similarity index (SSIM), and mean squared error (MSE). The definitions of these metrics are presented as follows:(7)$$MSE=\frac{1}{N}(y-G(x){)}^{2}$$
(8)$$PSNR=10\log_{10}^{\left(\frac{L^{2}}{\mathrm{MSE}}\right)}$$
(9)$$SSIM=\frac{\left(2\mu_{y}\mu_{G(x)}+c_{1})(2\sigma_{yG(x)}+c_{2}\right)}{\left(\mu_{y}^{2}+\mu_{G(x)}^{2}+c_{1})(\sigma_{y}^{2}+\sigma_{G(x)}^{2}+c_{2}\right)}$$
where y and G(x) are the ground truth and synthetic images, respectively. N represents the total number of pixels in each image slice. $\mu_{y}$, $\mu_{G(x)}$ and $\sigma_{y}$, $\sigma_{G(x)}$ are the mean and variance of the ground truth image and the synthesis image, and $\sigma_{yG(x)}$ is the covariance between y and G(x). $c_{1}=(k_{1}L{)}^{2}$ and $c_{2}=(k_{2}L{)}^{2}$ are two variables introduced to stabilize the SSIM division index with a weak denominator; here, the default setting of $k_{1}=0.01$ and $k_{2}=0.03$ are adopted, and L=255 is the dynamic range of the pixel intensity.

To evaluate the synthesis performance within the tumor region, we retrieved the clinical diagnosis report of the patients and selected twenty patients with confirmed tumors. According to the diagnosis report and tumor markers depicted by the radiologists, we manually drew three bonding boxes on the axial slice containing the tumor, i.e., the tumor region, the normal liver tissue region, and the background noise region. The tumor contrast-to-noise ratio (CNR) is defined as:(10)$$CNR=\frac{\left|\mu_{T}-\mu_{N}\right|}{s_{B}},$$
where $\mu_{T}$ and $\mu_{N}$ are the average values of the tumor region and the normal tissue region, respectively, and $\sigma_{B}$ is the standard deviation of the background ground noise region.

To evaluate the impact of image synthesis on liver tumor delineation, we invited 2 additional physicians (1 radiation oncologist attending with >20 years of experience and 1 senior medical resident trained by the same attending) to independently contour 21 liver tumors from 10 testing patients. The DICE coefficient (defined as the intersection over an average of two volumes) and Hausdorff distance (HD, measuring how far two volumes are from each other) were used as analysis metrics. Both physicians first performed tumor delineation on the real T1ce MRs. The DICE coefficients were calculated from the two physicians’ contours for each tumor. The average DICE (RadOnc7 vs. RadOnc 8) served as the baseline. The attending performed delineation on both the real T1ce MRs and synthetic T1ce MRs. The DICE (real vs. synth) coefficient and Hausdorff distance were calculated from volumes contoured on real and synthetic MRs for each tumor.

We performed tumor center shift analysis to further evaluate the potential effects of image-guided radiation therapy. This analysis extracted the tumor center of mass coordinates from real and synthetic volumes. The difference in the coordinates indicated the shifts in the superior–inferior (SI), right–left (RL), and anterior–posterior (AP) directions between real and synthetic tumor volumes.

## 4. Results

The synthesized T1ce was compared with the ground truth T1ce for thirty random patients. The overall performance is presented in Table 2. GRMM-GAN achieved a PSNR of 28.56 ± 0.87, an SSIM of 0.869 ± 0.028, and an MSE of 83.27 ± 15.42. GRMM-GAN outperformed all state-of-the-art multi-modal MR synthesis models and the single-modal pix2pix method. GRMM-GAN significantly improved all metrics over the comparison methods (*p*-value < 0.05). Figure 3 presents the synthesis results for one example patient. GRMM-GAN is shown to maintain rich details and textural consistency. The detail preservation is evident in the tumor region denoted by the red box.

In comparison, the other models, including both the single-input Pix2pix models and multi-input fusion models (LR-cGAN, Hi-Net, and MMgSN-Net), resulted in a substantial loss of anatomical details due to the lack of complementary feature mining and discrimination. The pix2pix methods with single input could not exploit the complementary information across different MR modalities. Although the pix2pix with T1pre input, as shown in Figure 3e, roughly predicted the tumor contour, it lost fine details, such as small vessels in the liver. The synthesis quality improved using LR-cGAN with added T2 input, as shown in Figure 3f, but the improvement on small vessel structures was modest. Hi-Net and MMgSN-Net more substantially improved fine structural preservation at the cost of compromising the contour integrity of the hypodense liver tumors. In contrast, our method, GRMM-GAN, retained the tumor integrity and preserved the high-contrast fine features, as shown in Figure 3i.

The Turing test results further substantiated the image quality preservation shown in Table 3. The average Turing test score from the 6 radiation oncologists was 52.33% ± 6.06, which is close to random guessing, indicating comparable visual quality between the synthetic and real images.

The contribution of each module was evaluated in an ablation test. The performance by different statistics metrics, PSNR, MSE, and SSIM, showed a consistent trend of additive value. Therefore, only MSE is described and discussed here for brevity. Figure 4 provides a visual evaluation of the ablation study. After removing the multi-scale and multi-smoothness (MMD) component, the average performance indicated by MSE increased from 83.27 ± 15.42 to 105.43 ± 16.15, highlighting the contribution of MMD in the discrimination of the low-quality synthesis. The exclusion of the GR model further increased the MSE to 121.65 ± 16.32, indicating the contribution of GR to structural preservation. Additionally, the average PSNR, SSIM, and MSE for the synthesized tumor region are 28.40, 0.856, and 88.71, respectively. The multiple performance evaluations show that the real and synthesized MR images for the specific tumor region are very similar, leading to the conclusion that the synthesized MR images could be a suitable surrogate for real contrast MR when the latter is unobtainable.

The average CNR of 20 patients for the real MR tumor region was 26.18 ± 21.13. In contrast, the average CNR for the synthesized MR tumor regions was 24.53 ± 20.08 with a *p*-value of 0.401 with no significant difference between the real and synthesized images for tumor analysis, suggesting similar tumor conspicuity provided by synthetic and real T1ce MR images.

RadOnc 7 and 8 achieved an average DICE (RadOnc7 vs. RadOnc 8) of 0.91 ± 0.02 from tumor volumes drawn on the real T1ce MRs. This result sets the inter-operator baseline in the real clinical setting. RadOnc 8 achieved an average DICE (real vs. synth) of 0.90 ± 0.04 and HD of 4.76 ± 1.82 mm. Only sub-millimeter tumor center shifts were observed in all three directions. The detailed tumor volume information is shown in Table 4.

## 5. Discussion

Contrast-enhanced MR images provide improved visualization of liver tumors, which is essential for MR-guided radiotherapy and SBRT. However, repeated injections of contrast may not be clinically viable. As an alternative, we synthesized contrast-enhanced MR images from multi-modality pre-contrast MR images using a novel GRMM-GAN model. The synthesized virtual contrast-enhanced MR images closely mimicked the ground truth contrast-enhanced images in quantitative image analysis and human expert Turing tests. The success indicates that the pre-contrast T1 and T2 images have a substantial predictive value for the post-contrast MR. The latent information in T1 and T2 images is intricate for human operators to appreciate but can be distilled via image synthesis into a format, e.g., post-contrast MR, familiar to human operators. Our technical innovations in building the GAN network overcame multiple challenges in existing image synthesis methods.

Several deep learning models have been developed for brain MR synthesis [34,48]. Compared with brain images, abdominal MR images are considerably more heterogeneous in soft tissue composition, MR relaxation properties, size, shape, and textures. The heterogeneity is further compounded by substantial motion in this anatomical region, leading to mismatches among different MR sequence acquisitions. Therefore, abdominal MR synthesis is a more challenging problem.

The challenge is highlighted by prior efforts using single-modal deep learning models for abdominal MR image synthesis [49,50], which fail to learn the salient knowledge across MR modalities, leading to a substantial loss in the synthesized image quality. The proposed GRMM-GAN is the first network to discover and fuse mutual complementary information, which markedly improves the synthesis performance, showing realistic contrast enhancement style translation, precise contours, and textures.

Our technical innovations are summarized as follows. First, a multi-modal fusion model was developed to exploit the salient and complementary information in multiple MR modalities. A sparse attention fusion module was investigated to distinguish the redundancy incurred. Second, we adopted the previously proposed image gradient regularization mechanism to avoid the loss of anatomical details in the GAN-based model and presented the novel multi-scale and multi-smoothness discrimination. In addition to these technical innovations, we prepared a valuable multi-modal abdominal database for contrast enhancement MR synthesis with patient data (165 scans from 61 patients) to support model development and validation.

The exclusion of the GR and MMD components decreases the average MSE to 105.43 ± 16.15 (*p*-value < 0.05) and 121.65 ± 16.32 (*p*-value < 0.05), respectively. Figure 4 provides an intuitive understanding of the essential contribution of each module. The absence of the MMD module degrades the discriminative ability of the network in over-smoothing and blurring effects, which can be visually observed in Figure 4c. The further removal of the GR module decreases the perception of the network to textural and structural details, as illustrated in Figure 4d. The specific tumor region synthesis evaluation also solidifies the proposed method in potential clinical application. The CNR and other performance evaluation metrics (PSRN, SSIM, and MSE) indicate that the salience and similarity of the synthesized specific tumor of interest are good enough compared with the real tumor data.

Similar to our GRMM-GAN model, the three multi-modal fusion generative methods, LR-cGAN, Hi-Net, and MMgSN-Net, generally performed better than the single modality synthesis model (pix2pix). The better performance could be attributed to the theory of multi-modal fusion to exploit the complementary information from different MR modalities. Specifically, MMgSN-Net was derived from Hi-Net and achieved the second-best overall performance boosted by the self-attention fusion module multi-modal feature learning. However, these methods still failed to discriminate the redundancy that existed accordingly, resulting in inferior synthesis quality with over-smoothing effects. For the single modal synthesis comparison, the pix2pix model yielded better performance with the T1pre as the input than with T2, likely because T1pre is more similar to the output T1ce modality. On the contrary, Li et al. [35] reported that the crucial knowledge from multi-modalities (e.g., T2) in revealing the hyper-intensity or pathological inhomogeneity characteristics could not be neglected.

Despite the excellent image synthesis quality and expert evaluation results, our work has the following limitations. We downsampled the original MR images from 256 × 256 to balance the image quality and computation complexity. This preprocessing could compromise the fidelity and precision of the synthesis results compared to the original resolution. A super-resolution [43,51] or interpolation [52,53] could be incorporated. Although quantitative metrics, such as PSNR, MSE, and SSIM, are suited for comparative image quality tests, they are not directly linked to more clinically relevant dosimetric measures for adaptive planning applications. We have performed an exploratory contouring study and showed satisfactory DICE coefficients calculated from real vs. synthetic MRs. In practice, the impact on the dosimetric endpoints will need to be evaluated for adaptive radiation therapy, which will require longitudinal MR images through the treatment course. Finally, our model is built upon diagnostic MR images; whether the network can be applied to MR-LINAC systems during or post-radiation MR synthesis will need to be tested.

## 6. Conclusions

This study developed a novel multi-modal contrast-enhanced liver MR image synthesis network (GRMM-GAN) capable of synthesizing highly realistic T1ce images that outperformed three state-of-the-art synthesis networks.

## Figures and Tables

**Figure 1 cancers-15-03544-f001:**
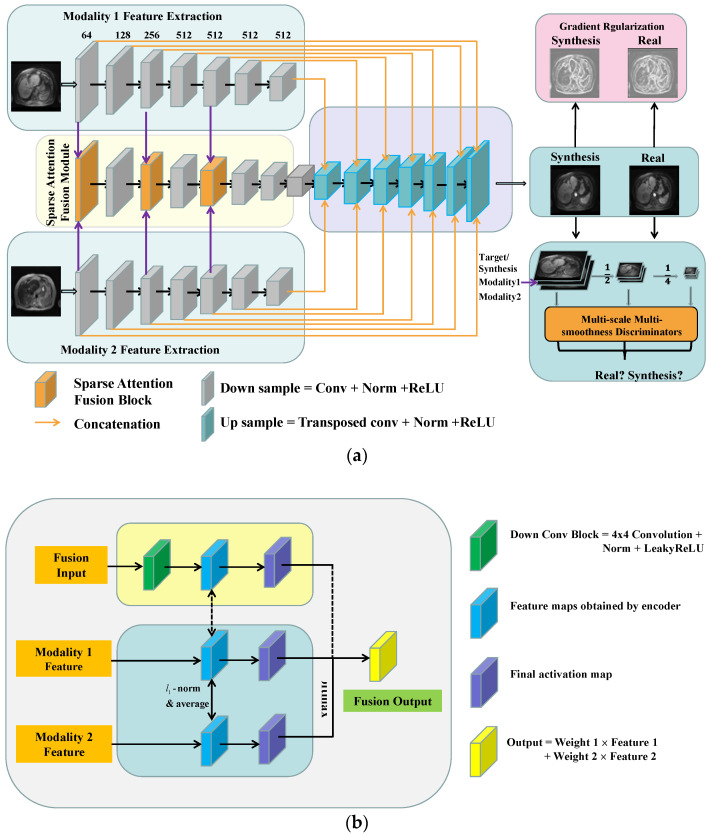
Framework of the proposed GR-MSSF GAN. (**a**) Network design and structure. (**b**) Sparse fusion module.

**Figure 2 cancers-15-03544-f002:**
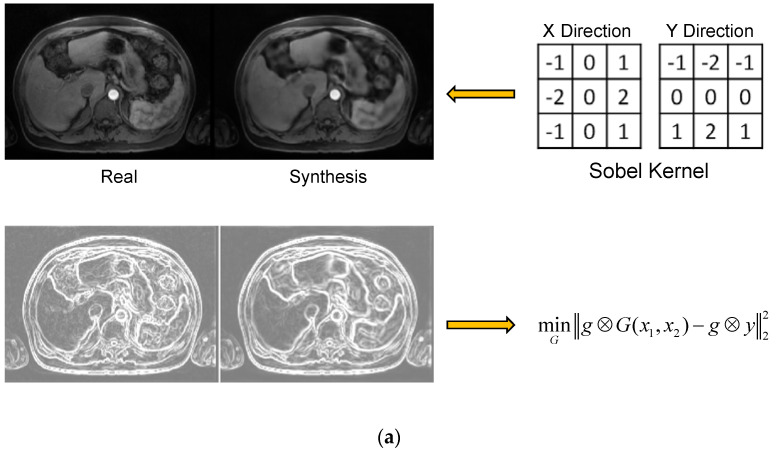
Illustration for the proposed gradient regularization and multi-scale multi-smoothness discrimination mechanism. (**a**) Gradient regularization (GR). (**b**) Multi-scale multi-smoothness discriminators (MMD).

**Figure 3 cancers-15-03544-f003:**
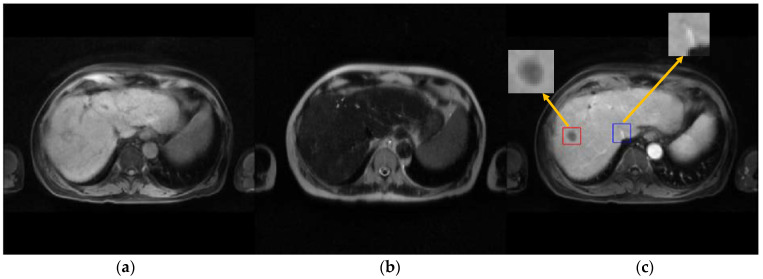
Synthesis example. (**a**) Input modality, T1pre. (**b**) Input modality, T2. (**c**) Target image, real T1ce. (**d**) T1ce synthesis, Pix2pix (T2). (**e**) T1ce synthesis, Pix2pix (T1pre). (**f**) T1ce synthesis, LR-cGAN. (**g**) T1ce synthesis, Hi-Net. (**h**) T1ce synthesis, MMgSN-Net. (**i**) T1ce synthesis, GRMM-GAN.

**Figure 4 cancers-15-03544-f004:**
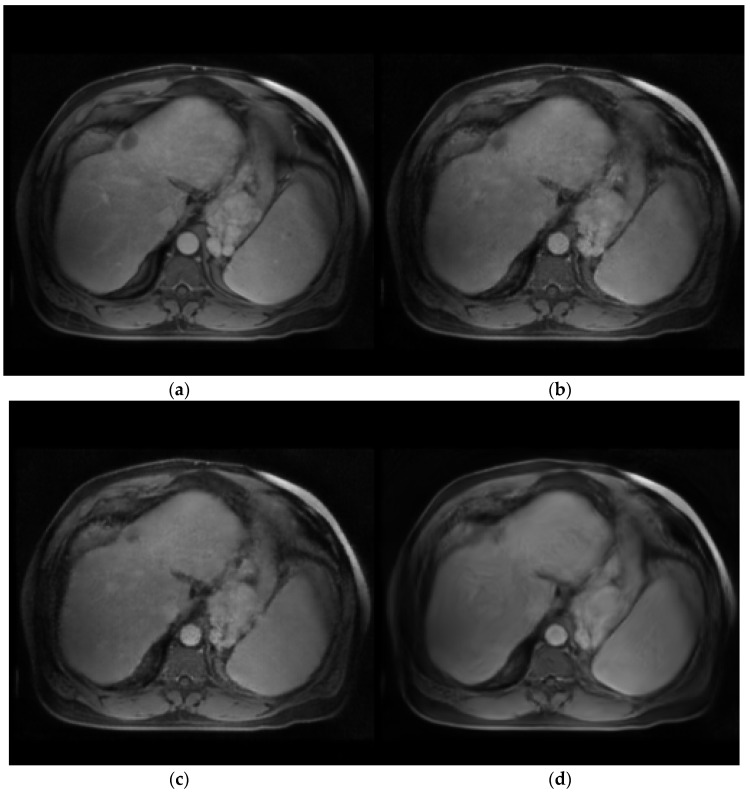
Visual examples for ablation study. (**a**) Target image, real T1ce. (**b**) GR+MMD. (**c**) Only GR, no MMD. (**d**) No GR, no MMD.

**Table 1 cancers-15-03544-t001:** Patient Characteristics.

Characteristic	Value
No. of patients	61 (Male: 45, Female: 16)
Age, median (range)	62 (37–83)
No. of studies	165
Type of liver cancer	Cholangiocaracinoma: 1
	Colon :5
	Colorectal: 1
	Esophageal adenoca: 1
	Gstric:1
	HCC:45
	Rectal:6
	Sigmoid adenocarcinoma:1
Stage at diagnosis	IA:4
	IB: 13
	II: 25
	IIIA: 2
	IIIB: 1
	IV: 16
Primary vs. metastatic	Primary: 44
	Metastatic: 16
	Both: 1
Average no. of liver tumors for the ten selected testing patients	2.1 (1–4)

**Table 2 cancers-15-03544-t002:** Performance Comparison of the Proposed GR-MMSF GAN and State-of-The-Art Comparisons in Contrast-enhanced Liver MR Synthesis and Ablation Study of the Proposed Method.

Methods	PSNR	SSIM	MSE
Pix2pix (T2) (20)	24.45 ± 1.33	0.786 ± 0.035	213.51 ± 29.81
Pix2pix (T1pre) (20)	24.82 ± 1.42	0.795 ± 0.039	192.32 ± 25.69
LR-cGAN (33)	25.91 ± 1.25	0.813 ± 0.032	141.48 ± 20.37
Hi-Net (34)	27.28 ± 1.26	0.836 ± 0.036	110.86 ± 21.04
MMgSN-Net (35)	28.04 ± 0.93	0.851 ± 0.033	98.43 ± 21.16
GR-MMSF GAN (proposed)	**28.56 ± 0.87**	**0.869 ± 0.028**	**83.27 ± 15.42**
Only GR, no MMD	27.61 ± 1.06	0.838 ± 0.031	105.43 ± 16.15
No GR, no MMD	26.84 ± 1.19	0.820 ± 0.034	121.65 ± 16.32

**Table 3 cancers-15-03544-t003:** Turing Test Results from Six Radiation Oncologists.

Radiation Oncologist	Evaluation	Results	Percentage (Correct)
1	Correct	55	55%
	Incorrect	45	
2	Correct	56	56%
	Incorrect	44	
3	Correct	42	42%
	Incorrect	58	
4	Correct	59	59%
	Incorrect	41	
5	Correct	53	43%
	Incorrect	47	
6	Correct	49	49%
	Incorrect	51	
Average			52.3%

**Table 4 cancers-15-03544-t004:** Tumor contouring evaluation on ten patients (with 21 lesions) from two radiation oncologists.

Radiation Oncologist	Average (Range) Volume (cc)	SI (mm)	RL (mm)	AP (mm)
7 (on real T1ce)	30.8 (1.2–233.7)	NA		
8 (on real T1ce)	29.4 (1.1–238.4)	0.67	0.41	0.39
8 (on synthetic T1ce)	28.6 (1.1–245.0)			

## Data Availability

Research data are stored in an institutional repository and will be shared with the corresponding author upon request.

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
