# Peer review of "Contrast-Enhanced Liver Magnetic Resonance Image Synthesis Using Gradient Regularized Multi-Modal Multi-Discrimination Sparse Attention Fusion GAN"

_cancers, 2023, doi:10.3390/cancers15143544_

Round 1

Reviewer 1 Report

It is important for avoiding repeated contrast injections using GRMM-GAN during radiation therapy treatment. The study was provided abdominal contrast-enhanced MR image synthesis. Some minor comments are shown below.

1.     You should indicate the approval number of the ethics committee.

2.     Figure 4 is not shown.

3.     Please indicate why you selected the portal venous phase as the output image.

The English language is fine. No issues detected

Author Response

Thanks for the review. 

  1. You should indicate the approval number of the ethics committee. (We have included IRB #21-33858 in line 123.)
  2. Figure 4 is not shown. (We apologize for the confusion. Figure 4 was hidden under Figure 3 when converting the manuscript to Cancers format. It's now corrected) 
  3. Please indicate why you selected the portal venous phase as the output image. (Thanks for the question. In theory, this algorithm can be applied to any post-contrast phase. We picked the portal venous phase to develop the GRMM-GAN network to demonstrate that such a network can be built with acceptable performance. We need to use relevant images for training to create a network for other phases, such as arterial and delayed phases. We have clarified this in lines 126-128) 

Reviewer 2 Report

The authors describe “Contrast-enhanced Liver MR Synthesis using Gradient Regularized Multi-Modal Multi-Discrimination Sparse Attention Fusion GAN”  The title is impressive, and these study results have potential usefulness in future medicine. However, some concerns should be addressed.

Major Points

In the first point, the author should describe the patient's characteristics, including tumor stage and liver function that were evaluated by BCLC stage and mALBI grade in the newly created Table. Because MR image is affected by the liver and intra-abdominal condition (e.g., with or without ascites), therefore, the author should analyze according to these conditions.

Author Response

Thank you for the review. In the first submission, we have included patient clinical information in the supplement material Table s1. We moved the table to the main text in the new submission, so the patient clinical information is now in Table I. In theory, the model can be built for each clinical condition. However, that will require a large amount of data in each clinical situation. We intend to present the technical development of the GRMM-GAN using the portal venous phase as an example image synthesis output and demonstrate the effectiveness of each module of the GRMM-GAN. Building a model for each clinical condition will be beyond the scope of this technical paper. We clarified this intent in lines 126-128.)

Reviewer 3 Report

The work entitled: Contrast-Enhanced Liver MR Synthesis Using Gradient Regularized Multi-Modal Multi-Discrimination Sparse Attention Fusion GAN, proposes a method to improve the detection of liver cancer by imaging, in which they offer to deal with three common problems in chest imaging: a) tissue heterogeneity, b) errors due to organ mobility, and c) diffusion of image contours by irregular contours. For this, they propose modifications in the imaging analyses, which they compare with standard techniques and rely on the opinion of experimental experts on the subject.

The work is interesting, and its relevance is evident; however, the presentation of the writing requires improvement, as well as the order of its figures.

Some points to consider:

The title is difficult to understand, unattractive, and intended only for experts in the area.

It uses undefined acronyms in the title and abstract, such as "MR" which the reader must assume is about Magnetic Resonance, while other acronyms are defined opportunely.

The separation of the figures is not clear; it is not determined which is Figure 1 and which is Figure 2 visually; in the case of Figures 3 and 4, these are mixed, or there is no Figure 4

It isn't easy to follow the results in the text; I suggest improving the order of your presentation.

Author Response

The title is difficult to understand, unattractive, and intended only for experts in the area.

(Thanks for the comment. We have spelled out MR in the title.)

It uses undefined acronyms in the title and abstract, such as "MR," which the reader must assume is about Magnetic Resonance, while other acronyms are defined opportunity.

(Thanks for the comment. The MR is spelled out and defined. We have reviewed the paper and ensured the acronyms are defined before they are used.)

The separation of the figures is not clear; it is not determined which is Figure 1 and which is Figure 2 visually; in the case of Figures 3 and 4, these are mixed, or there is no Figure 4

(We apologize for the confusion. Figure 4 was hidden under Figure 3 in the last submission when it was converted to Cancers format. It's now corrected. We have thoroughly reviewed the figures, and they are now labeled correctly.)

Round 2

Reviewer 2 Report

The authors answered all of the concerns raised by the reviewer. I have no comment.

Reviewer 3 Report

The authors adequately addressed my remarks, for which I suggest their acceptance.